# Targeted Nutritional Intervention for Patients with Mild Cognitive Impairment: The Cognitive impAiRmEnt Study (CARES) Trial 1

**DOI:** 10.3390/jpm10020043

**Published:** 2020-05-25

**Authors:** Rebecca Power, John M. Nolan, Alfonso Prado-Cabrero, Robert Coen, Warren Roche, Tommy Power, Alan N. Howard, Ríona Mulcahy

**Affiliations:** 1Nutrition Research Centre Ireland, School of Health Sciences, Carriganore House, Waterford Institute of Technology West Campus, X91 K0EK Waterford, Ireland; jmnolan@wit.ie (J.M.N.); aprado-cabrero@wit.ie (A.P.-C.); warren.roche@ucdconnect.ie (W.R.); tbpower@wit.ie (T.P.); 2Mercer’s Institute for Research on Ageing, St. James’s Hospital, D08 NHY1 Dublin, Ireland; rcoen@stjames.ie; 3Howard Foundation, 7 Marfleet Close, Great Shelford, Cambridge CB22 5LA, UK; alan.howard@howard-foundation.com; 4Age-Related Care Unit, Health Service Executive, University Hospital Waterford, Dunmore Road, X91 ER8E Waterford, Ireland; 5Royal College of Surgeons Ireland, 123 Stephen’s Green, Saint Peter’s, D02 YN77 Dublin, Ireland

**Keywords:** mild cognitive impairment, nutrition, omega-3 fatty acids, antioxidant, carotenoids, vitamin E, cognition, episodic memory, older adults, ageing

## Abstract

Omega-3 fatty acids (ω-3FAs), carotenoids, and vitamin E are important constituents of a healthy diet. While they are present in brain tissue, studies have shown that these key nutrients are depleted in individuals with mild cognitive impairment (MCI) in comparison to cognitively healthy individuals. Therefore, it is likely that these individuals will benefit from targeted nutritional intervention, given that poor nutrition is one of the many modifiable risk factors for MCI. Evidence to date suggests that these nutritional compounds can work independently to optimize the neurocognitive environment, primarily due to their antioxidant and anti-inflammatory properties. To date, however, no interventional studies have examined the potential synergistic effects of a combination of ω-3FAs, carotenoids and vitamin E on the cognitive function of patients with MCI. Individuals with clinically confirmed MCI consumed an ω-3FA plus carotenoid plus vitamin E formulation or placebo for 12 months. Cognitive performance was determined from tasks that assessed global cognition and episodic memory. Ω-3FAs, carotenoids, and vitamin E were measured in blood. Carotenoid concentrations were also measured in tissue (skin and retina). Individuals consuming the active intervention (*n* = 6; median [IQR] age 73.5 [69.5–80.5] years; 50% female) exhibited statistically significant improvements (*p* < 0.05, for all) in tissue carotenoid concentrations, and carotenoid and ω-3FA concentrations in blood. Trends in improvements in episodic memory and global cognition were also observed in this group. In contrast, the placebo group (*n* = 7; median [IQR] 72 (69.5–75.5) years; 89% female) remained unchanged or worsened for all measurements (*p* > 0.05). Despite a small sample size, this exploratory study is the first of its kind to identify trends in improved cognitive performance in individuals with MCI following supplementation with ω-3FAs, carotenoids, and vitamin E.

## 1. Introduction

Given the growing social and economic burden of cognitive decline on society, emphasis is being placed on preventative strategies to delay the onset and reduce the risk of developing dementia, with particular focus on Alzheimer’s disease (AD) as it is the most common form of dementia. Mild cognitive impairment (MCI) is often a transitional phase between the cognitive changes that one expects as one ages and very early dementia. It is recognized as a deterioration in cognitive function that exceeds what is anticipated for an individual based on their age and education level. Importantly, these changes in cognition are not significant enough to impact an individual’s independence or ability to perform activities of daily living [1]. MCI is difficult to diagnose, and its prognosis is notoriously unpredictable. While the mortality rate is higher in MCI patients in comparison to cognitively healthy individuals [2], it is comparable to dementia mortality rates [3]. Although MCI is a risk factor for AD (with MCI to dementia conversion rates estimated at 3%–15% annually [4]), it is important to note that some individuals with the condition remain stable and do not progress while others may improve (i.e., revert to a cognitively intact state) upon follow-up assessment. This reversion phenomenon is an inherent feature of MCI and may be explained by the heterogeneity of the condition. While reverting from MCI to a cognitively intact state seems like a positive outcome, importantly, a number of studies have shown that these individuals are in fact at a greater risk of cognitive decline in the future [5,6]. Thus, due to the increased risk of mortality and progression to AD, MCI is an important public health concern.

Despite its complex and dynamic nature, MCI offers a window of opportunity to examine the potential of preventative strategies for modifying or delaying disease progression and improving cognitive outcomes. Given that many risk factors (e.g., vascular disease, diabetes, smoking, physical inactivity, social isolation [7,8,9]) for MCI are modifiable, shifting focus towards preventative strategies seems prudent. Specifically, there is a growing body of evidence to suggest that good nutrition is important for cognitive performance [10,11,12,13,14], and is associated with a reduced risk of MCI and AD [15,16,17,18]. Omega-3 fatty acids (ω-3FAs), xanthophyll carotenoids (oxygen-containing, plant-based pigments), and vitamin E are important constituents of a healthy diet. While these specific nutrients are present in brain tissue [19,20,21], studies have shown that they are depleted in individuals with MCI and AD in comparison to cognitively healthy individuals [22,23,24]. Therefore, it is likely that specific population groups (e.g., individuals with MCI, very early-stage AD or individuals with a low ω-3FA or carotenoid index) will benefit from targeted nutritional intervention. Indeed, observational [20,25,26,27,28,29] and interventional [30,31,32,33] evidence to date suggests that these nutritional compounds can work independently to optimize the neurocognitive environment [34], primarily due to their antioxidant and anti-inflammatory properties. Interestingly, previous exploratory work has shown that a combination of ω-3FAs and xanthophyll carotenoids can work synergistically to improve cognition in older women [35], and maintain function and quality of life in AD patients [36]. To date, however, no interventional studies have examined the potential synergistic effects of a combination of ω-3FAs, xanthophyll carotenoids, and vitamin E on the cognitive health and function of patients with MCI.

The present study, the Cognitive impAiRmEnt Study (CARES), was designed to investigate the impact of targeted nutritional intervention with ω-3FAs, xanthophyll carotenoids, and vitamin E on cognitive function among individuals with MCI. CARES was a parallel group, double-blind, placebo-controlled, randomized clinical trial studying two populations of interest. The first arm of the trial (CARES Trial 1) examined the impact of targeted nutritional supplementation on cognitive function in individuals with MCI, while the second arm of the trial (CARES Trial 2) investigated the impact of targeted nutritional supplementation on cognitive function in cognitively heathy older adults (≥65 years). Herein, exploratory work from CARES Trial 1 is presented and discussed.

## 2. Materials and Methods

### 2.1. Study Design

CARES Trial 1 investigated the impact of 12-month supplementation with ω-3FAs, xanthophyll carotenoids, and vitamin E on cognitive function in individuals with MCI. Individuals were initially identified as potentially suitable for enrolment based on a medical assessment performed by consultant geriatricians and psychiatrists of old age in the South-East catchment area of Ireland. Both amnestic and non-amnestic MCI were included. MCI sub-type classification was not performed. A diagnosis of MCI was based on published criteria [37,38]. Specific eligibility criteria included: self or family member reported memory loss; fulfilled criteria for minimal cognitive impairment; functionally independent in activities of daily living; ≥65 years of age; no rapidly progressive or fluctuating symptoms of memory loss; no established diagnosis of early dementia (consumption of cognitive enhancement therapy such as cholinesterase inhibitors or N-methyl-D-aspartate receptor antagonists); no stroke disease (clinical stroke or stroke on CTB); no depression (under active review); no psychiatric illness (under active review of psychotropic medications); no glaucoma (acute angle); no consumption of carotenoid or fish/cod liver oil supplements; and no fish allergy.

### 2.2. MCI Screening

Prior to enrolment, all individuals who expressed an interest in participating in the trial completed a screening assessment to confirm eligibility. This included assessing cognitive function using the Repeatable Battery for the Assessment of Neurological Status (RBANS) Record Form A and the Montreal Cognitive Assessment (MoCA) version 7.1. Level of functional ability was assessed using the Bristol Activities of Daily Living Scale (BADLS) and the Alzheimer’s Questionnaire (AQ). A brief description of each of these assessments is provided below. In the event where an informant was not present during the assessment, a family member or carer was contacted via telephone to complete functional ability assessments. In circumstances where no informant was available, the researcher administered the questionnaires to the patient. Individuals who fulfilled the criteria for each cognitive and functional assessment were invited to participate in the clinical trial. Individuals with borderline scores from the screening assessments were referred to a consensus panel (via email or conference call) consisting of one consultant geriatrician, one psychiatrist of old age and one clinical neuropsychologist for assessment of eligibility [39]. Eligible individuals were then invited to enroll into the study (see Figure 1). Prior to enrolment, written informed consent was obtained from all individuals. Ethical approval was granted by the Research Ethics Committees of the Waterford Institute of Technology and University Hospital Waterford in Waterford, Ireland in December 2015. CARES (trial registration number: ISRCTN10431469) adhered to the tenets of the Declaration of Helsinki and followed the full code of ethics with respect to recruitment, testing and general data protection regulations as set out by the European Parliament and Council of the European Union.

Eligible individuals were randomized to either the active intervention (now commercially known as Memory Health) containing 1 g of fish oil (of which 430 mg docosahexaenoic acid [DHA] and 90 mg eicosapentaenoic acid [EPA]), the xanthophyll carotenoids lutein (L) (10 mg), *meso*-zeaxanthin (MZ) (10 mg) and zeaxanthin (Z) (2 mg), and 15 mg of vitamin E (α-tocopherol), or placebo (sunflower oil) intervention group. These doses were provided via two oval-size capsules. Each capsule contained equal quantities of fish oil, carotenoids and vitamin E (see Table A1 in the Appendix A section). Carotenoid and vitamin E concentrations were manufactured by Industrial Orgánica (Monterrey, Mexico), while fish oil concentrations were manufactured by Epax (Ålesund, Norway; product number: EPAX1050TG/N non-tuna). The complete formula composition and the concentration of fatty acids of total lipids are available in the Appendix A section (Table A1 and Table A2, respectively in Appendix A). Individuals were instructed to consume two capsules per day with a meal. Frequent phone calls were made to ensure compliance. Tablet counting was also performed at follow-up. Study visits were conducted at baseline and 12 months at a single site (Nutrition Research Centre Ireland [NRCI]). Intervention randomization was performed by an electronic trial management system (Trial Controller) designed by our research group (NRCI). This administration system was also used to document patient information (name and contact details), support the organization and management of capsules required for the clinical trial and assist with the scheduling of study visits. The primary outcome measure of CARES Trial 1 was change in cognitive function. Secondary outcome measures included change in the following variables: macular pigment optical volume (MPOV); visual function; serum xanthophyll carotenoid concentrations (L, Z and MZ); serum vitamin E concentrations (α-tocopherol); and plasma ω-3FA concentrations (EPA and DHA). Development of AD was also recorded.

### 2.3. Assessing Cognitive Function

#### 2.3.1. Global Cognition

The MoCA was used at the screening stage to assess global cognition. It is a short (10-min) cognitive screening tool with high sensitivity and specificity for detecting MCI [40]. Thirty items assess multiple cognitive domains including visuospatial abilities, executive function, phonemic fluency, attention, immediate and delayed recall, language and orientation. The RBANS was used to measure global cognition at screening and at 12-month follow-up visits. Five domains of cognition (immediate memory, visuospatial ability, language, attention and delayed memory) were assessed using 12 sub-tests. A composite or “total index/scale score” was also computed. The RBANS takes approximately 30 min to administer and is a core diagnostic tool for detecting and characterizing dementia [41]. The RBANS yields index standard scores that are based on the raw scores of each subtest. RBANS index scores are metrically scaled, with a mean of 100 and a standard deviation (SD) of 15 for each age group. A score of 100 on any of these measures equates to the average performance of individuals of similar age. Scores of 85 and 115 correspond to 1 SD below and above the mean, respectively, while scores of 70 and 130 are 2 SDs below and above the mean. Approximately 68% of all examinees score between 85 and 115, circa 95% score in the 70 to 130 range, and nearly all examinees obtain scores between 55 and 143 [42]. In the present study, scores < 78 and between 19 and 25 for the RBANS and the MoCA, respectively, were desirable for enrolment.

#### 2.3.2. Specific Cognitive Domains

Additional assessments of specific cognitive domains were performed using the Cambridge neuropsychological test automated battery (CANTAB) Connect Research Software (Cambridge Cognition, Cambridge, UK) [43]. This computerized software program was performed on an iPad and required a finger-operated response. This technology has been previously tested and validated in older adult population groups [44]. The CANTAB protocol [45] was followed in the administration of the test battery and was used to assess comprehension, executive function (working memory), attention (reaction time) and episodic memory at baseline and follow-up visits. Table 1 provides an overview of the CANTAB tests performed.

Spatial memory was also assessed at the screening stage only using the 4 mountains test (4MT) [46]. Using a delayed match-to-sample paradigm, memory for the topographical layout of 4 mountains within a computer-generated landscape is tested. Individuals were asked to recall the spatial configuration of a total of 15 sets of computer-generated landscapes from a shifted viewpoint, which is designed to reflect the role of the hippocampus in spatial cognition. This computerized assessment was performed on an iPad and required a finger operated response. The test takes approximately 20 min to complete and has been used previously among individuals with MCI and AD [47]. A study by Moodley and colleagues [48] suggested that a total 4MT score of ≤8 was associated with 100% sensitivity and 90% specificity for detecting early AD when tested in a UK population, and associated with 100% sensitivity and 50% specificity for detection of MCI and AD when tested in an Italian population group.

### 2.4. Assessing Functional Ability

The BADLS is an informant-based, 20-item questionnaire designed to measure the ability of an individual with dementia to carry out activities of daily living such as washing, dressing, preparing food and using transportation [49,50]. It is sensitive to changes in dementia and is regularly used as an outcome measure in clinical trials, where it is world leading as a dementia-specific measure. This outcome is among those recommended by a consensus recommendation of outcome scales for non-drug interventional studies in dementia [51]. A higher score was desirable for this assessment. The AQ is an informant-based screening tool used to detect cognitive impairment. It is regarded as a time efficient and sensitive measure for detecting MCI using structured interview-based questions. The AQ consists of 21 yes/no questions in a weighted format relevant to five different domains: memory, orientation, functional ability, visuospatial and language. The total score is calculated by summing the number of items with a “yes” response. Clinical symptoms known to be highly predictive of AD are given a greater weight in the total score. A score between 5 and 14 points was desirable for this assessment. The AQ has been previously validated and has shown high sensitivity and specificity for detecting MCI [52,53].

### 2.5. Assessing Nutritional Status

#### 2.5.1. Macular Pigment

The carotenoids L, Z and MZ are preferentially concentrated in the central retina (macula lutea, which is part of the central nervous system) where they are collectively referred to as macular pigment (MP). MP was measured by dual-wavelength autofluorescence (AF) using the Spectralis HRA+OCT MultiColor (Heidelberg Engineering GmbH, Heidelberg, Germany). Pupillary dilation was performed prior to measurement and patient details were entered into the Heidelberg Eye Explorer (HEYEX version 1.7.1.0) software. Dual-wavelength AF in this device uses two excitation wavelengths; one that is well absorbed by MP (486 nm, blue) and one that is not (518 nm, green) [54]. The following acquisition parameters were used: high speed scan resolution, two seconds cyclic buffer size, internal fixation, 30-s movie and manual brightness control. Alignment, focus and illumination were first adjusted in infrared mode. Once the image was evenly illuminated, the laser mode was switched from infrared to blue plus green laser light AF. Using the HEYEX software, the movie images were aligned and averaged, and a MP density map was created. MPOV, calculated as MP average times the area under the curve out to 7° eccentricity [55], is reported here. This system has recently been validated by our research center [56].

#### 2.5.2. Skin Carotenoid Score

Carotenoid concentrations were also measured using the Pharmanex^®^ BioPhotonic Scanner (Salt Lake City, UT, USA). This scanner measures carotenoid levels in human tissue at the skin surface using optical signals (resonant Raman spectroscopy). These signals identify the unique molecular structure of carotenoids, allowing their measurement without interference by other molecular substances. The individual was asked to place a specific point (between the maximal and distal palmar creases, directly below the fifth finger) of their right hand (previously cleaned with hand sanitizer) in front of the scanner’s low-energy blue light for 30 s. From this, a skin carotenoid score (SCS) was generated. This provided an indication of the individual’s overall antioxidant levels. This was repeated twice more, and an average score was calculated. Based on this result, an individual’s score can be classified into three ranges: 0–29,000 = low; 30,000–49,000 = normal; ≥50,000 = high. This technology is safe and has been previously validated [57].

### 2.6. Biochemical Analysis of Serum Xanthophyll Carotenoids and Vitamin E

#### 2.6.1. Serum Extraction

Non-fasting blood samples were collected at each study visit by standard venipuncture techniques. SST II Advance blood collection tubes (8.5 mL) were inverted at least 5 times to ensure thorough mixing of the silica clot activator. The blood samples were left to clot for 30 min at room temperature and then centrifuged at room temperature at 725 g for 10 min in a Gruppe GC12 centrifuge (Desaga Sarstedt, UK) to separate the serum from the whole blood. Following centrifugation, serum was transferred to light-resistant microtubes and stored at circa −80 °C until extraction. Xanthophyll carotenoids and α-tocopherol were extracted from serum samples as previously described [58] and analyzed by high performance liquid chromatography (HPLC).

#### 2.6.2. Lutein, Zeaxanthin and α-Tocopherol Quantification (Assay 1)

The chromatographic analysis of carotenoids and α-tocopherol was performed on an Agilent 1260 Series HPLC (Agilent Technologies Limited, Santa Clara, CA, USA) equipped with a quaternary pump, autosampler, thermostat column compartment and a photodiode array detector monitoring a wavelength of 450 nm for serum carotenoids and 292 nm for α-tocopherol and the internal standard (IS) α-tocopheryl acetate. The dried samples were reconstituted in 0.2 mL of Methanol:MTBE (9:1, *v*/*v*), vortexed at the lowest setting for 1 min and pipetted into HPLC vials containing 0.35 mL glass inserts. 0.1 milliliters of each sample was injected in a C30 carotenoid column (250 × 4.6 mm i.d., 3 μm; YMC Europe, Dinslaken, Germany) with a guard column of the same chemistry. HPLC mobile phase A consisted of methanol:MTBE:water (83:15:2, *v*/*v*), and mobile phase B consisted of methanol:MTBE:water (8:90:2, *v*/*v*), both with 0.1% BHT. At a flow rate of 1 mL min^−1^, the gradient initiated at 5% solvent B and increased to 20% in the first 12 min, to 55% over the next 8 min and to 95% over the next 7 min. From 27–30 min, solvent B was held at 95%, and then resumed to initial setting within 3 min. Separations were carried out at 16 °C. Total Z from each sample was automatically collected by the fraction collector in amber eppendorfs.

#### 2.6.3. *Meso*-Zeaxanthin Quantification (Assay 2)

Total Z collected in HPLC system 1 was dried in a vacuum centrifuge and re-suspended in 0.2 mL of hexane:isopropanol (90:10, *v*/*v*). 0.1 milliliters of the sample was analyzed on another Agilent 1260 Series HPLC system equipped with a Diode Array Detector, binary pump, degasser, thermostatically controlled column compartment and thermostatically controlled high-performance autosampler. The column used for the separation of the stereoisomers of Z was a Daicel Chiralpak IA-3 column, composed of amylose tris (3,5-dimethylphenylcarbamate) bonded to 3 mm silica gel (250 × 4.6 mm i.d.; Chiral Technologies Europe, Cedex, France). The column was protected with a guard column containing a guard cartridge with the same chemistry of the column. Isocratic elution was performed with hexane:isopropanol (90:10, *v*/*v*) at a flow rate of 0.5 mL min^−1^. The column temperature was set at 20 °C.

Quantification was performed by constructing a calibration line for each xanthophyll carotenoid analyzed and for α-tocopherol. For each compound of interest, at least five calibration standards were quantified using a UV‒Vis spectrophotometer UVmini-1240 (Shimadzu, Kyoto, Japan) with the appropriate molar extinction coefficient (see Appendix A
Table A3). These calibrators were analyzed using HPLC system 1 in triplicate, whereas the calibrator of lowest concentration was injected 10 times in order to experimentally calculate the lower limit of quantification (LLOQ) for the compound. The upper limit of quantification (ULOQ) was allocated as the calibrator of highest concentration of each calibration curve analyzed in triplicate (see Appendix A
Table A4). Where possible, subject samples that displayed an area in the HPLC chromatogram below the LLOQ or above the ULOQ were re-analyzed in order to obtain an area within the range of the calibration line. If after re-analysis the area of the analyte of interest remained below the LLOQ, this analyte in the subject was marked as ‘below LLOQ’. In order to determine the efficiency and precision of the xanthophyll carotenoid and α-tocopherol quantification methodology, analyte recovery analysis, precision analysis and trueness of sample recovery were performed. Details of this analysis are outlined in the Appendix A section.

### 2.7. Biochemical Analysis of Plasma Omega-3 Fatty Acids

#### 2.7.1. Plasma Extraction

Lithium heparin blood collection tubes (6 mL) were inverted 8–10 times to ensure thorough mixing and were centrifuged at 4 °C at 3000 rpm for 20 min in a 3–18 K centrifuge (Sigma-Aldrich, St. Louis, MO, USA) to separate red blood cells and plasma. The time of blood collection and time of separation did not exceed 2 h. Following centrifugation, all samples were transferred to light-resistant microtubes and stored at circa −80 °C until the time of analysis. Plasma ω-3FA analysis was performed by gas chromatography (GC). Fatty acid methyl esters (FAME) were prepared as previously described [59]. Briefly, 50 μL of plasma were spiked with 20 μL of 2 mg/mL methyl tricosanoate (Larodan, Solna, Sweden) to assess FAME recovery and saponified with 2 mL of freshly prepared methanolic KOH 0.4 M during 10 min with gentle vortexing at room temperature. The samples were extracted three times with 2 mL of hexane and the combined extracts were dried in a vacuum centrifuge. The pellets were esterified with 2 mL of freshly prepared 5% methanolic sulfuric acid (*v*/*v*) at 80 °C for 30 min in a thermo-block. The FAME produced were extracted three times with 2 mL of hexane and dried in the vacuum centrifuge. The samples were resuspended in 0.4 mL of hexane containing 0.1 mg/mL of methyl heneicosanoate (Larodan) to assess the matrix effect and prepared for GC analysis. Methyl tricosanoate and methyl heneicosanoate 0.1 mg/mL were injected in triplicate to assess recovery and matrix effect, respectively.

#### 2.7.2. DHA and EPA Quantification

FAME were quantified by GC coupled to flame ionization detector (GC-FID) with an Agilent 7890B Gas Chromatographer, using a Thermo 260M142P column (cyanopropylphenyl-based phase, 30 m length, 0.25 mm inner diameter and 0.25 µm film thickness). Nitrogen was used as the carrier gas with a flow rate of 1.5 mL/min and an electronic pressure control at 20.8 psi. Temperature ramp started at 140 °C and was held for 1 min, then followed by an increase of 6 °C min^−1^ until 210 °C, an increase of 2.5 °C min^−1^ until 230 °C and finally an increase of 10 °C min^−1^ until 240 °C, which was maintained for 5 min. Total run time was 26.7 min, with post run temperature at 50 °C and maximum temperature at 250 °C. FAME were identified by comparison with the authentic standard Mixture ME 1220 (Larodan). For FAME quantification, an RF was calculated as follows: a calibration line for methyl docosanoate, methyl undecanoate, methyl heptadecanoate, methyl heneicosanoate, methyl tricosanoate and EPA was prepared with a concentration range of 0.0025–0.5 mg/mL and analyzed in the GC. The resulting calibration lines were forced to pass through the origin of the axes, and the resulting slopes were averaged. The resulting RF was 0.1572 ± 0.0125.

### 2.8. Additional Biochemical Analysis

Serum and plasma samples were also collected to measure sodium, potassium, chloride, creatinine, total cholesterol, triglycerides, HDL, LDL, folate, vitamin B12, homocysteine, C-reactive protein, thyroid stimulating hormone, and free T4 (see Table A9 in the Appendix A section). One K2EDTA blood collection tube (3 mL) was also used for whole blood analysis. The sample was inverted 8–10 times and refrigerated at 4 °C until sample collection (2–24 h later). This additional analysis was performed by an accredited medical testing service provider (Eurofins Biomnis, Dublin, Ireland).

### 2.9. Demographic, Health and Lifestyle Data

Demographic, health and lifestyle data, medical history and medication use were recorded via questionnaire. Height and weight measurements were recorded to calculate body mass index (BMI) (kg/m^2^). Smoking status was classified into never (smoked < 100 cigarettes in lifetime), past (smoked ≥ 100 cigarettes in lifetime and none in the past year) or current (smoked ≥ 100 cigarettes in lifetime and at least 1 cigarette in the last year) smoker. Alcohol consumption was measured in unit intake per week. One unit of alcohol (10 mL) was the equivalent to one of the following: a single measure of spirits (ABV 37.5%); half a pint of average-strength (4%) lager; two-thirds of a 125 mL glass of average-strength (12%) wine; half a 175 mL glass of average-strength (12%) wine; a third of a 250 mL glass of average-strength (12%) wine. Color fundus photographs were taken to assess the presence of ocular pathology (Zeiss Visucam 200, Carl Zeiss Meditec AG, Jena, Germany).

### 2.10. Statistical Analysis

The statistical package IBM SPSS version 25 was used and the 5% significance level applied for all analyses. Given that data were not normally distributed, the small sample size and the presence of ranked data, a non-parametric approach was taken. Results were expressed as median (inter quartile range [IQR]) for all variables. Between-group differences (i.e., active versus placebo) were analyzed using Mann–Whitney U, Wilcoxon Signed Rank or Chi-square tests, as appropriate. The Mann–Whitney U test was also used to examine the significance of change in nutrition variables over time between active and placebo intervention groups. Significance values were not computed to examine change in cognition or vision variables over time between both groups due to a lack of statistical power and the small magnitude of change over time observed for these variables. As an alternative, the average percentage change per subject was reported. Of note, percentage change could not be calculated for some variables (e.g., episodic memory) as baseline values were recorded as 0. Thus, the average change per subject was reported.

## 3. Results

### 3.1. Baseline Results

Table 2 presents the baseline demographic, health and lifestyle data for active and placebo intervention groups. Table 3 presents the baseline cognitive function and functional ability for active and placebo intervention groups. Baseline variables were statistically comparable between both groups, with the exception of the number of between errors (*p* = 0.006) and total errors (*p* = 0.012) made at stage 8 of the SWM tasks, which were significantly higher in the active group. Additionally, creatinine levels were significantly higher in the active group at baseline (*p* = 0.008), but were within normal ranges (see Table A9 in the Appendix A section). No comprehension or sensorimotor difficulties were observed during the CANTAB assessment (see Materials and Methods section), as the motor screening task (MOT) latency assessment was completed by all individuals at baseline and follow-up. No adverse events were reported by individuals in the active or placebo intervention groups during the trial.

### 3.2. Observed Change in Nutritional Status

Table 4 summarizes the observed change in nutrition variables for both groups following the 12-month intervention period. Figure 2A–C illustrate the observed changes in MPOV, serum L concentrations and plasma DHA concentrations, respectively. Individuals in the active intervention group exhibited statistically significant improvements in MPOV (62% improvement versus 2% decline for active and placebo groups, respectively; *p* = 0.001) and SCS (79% improvement versus 2% improvement for active and placebo groups, respectively; *p* = 0.014) in comparison to individuals in the placebo group. In terms of biochemical response, individuals in the active intervention group exhibited statistically significant improvements in serum carotenoid concentrations of L and MZ, as well as statistically significant improvements in plasma concentrations of DHA (*p* < 0.05, for all) in comparison to individuals receiving placebo. Serum Z and plasma EPA levels increased in both groups; however, results were not statistically significant (*p* > 0.05, for all). A mixed response to vitamin E supplementation was observed in blood, where levels decreased (−4%) over 12 months in the active intervention group and increased slightly (+1%) in the placebo group.

### 3.3. Observed Change in Global Cognition

Table 5 shows trends in improvements (ranging from 6% to 18%) in global cognition (as per the RBANS assessment tool) in the active intervention group after 12 months. Specifically, trends in improvements were observed in the immediate memory, attention and delayed memory domains, as well as the total scale score. Minor declines were denoted in the visuospatial and language domains (both by 1%). Global cognition results were mixed in the placebo group. Immediate memory, visuospatial and attention domains of the RBANS remained unchanged while language, delayed memory and total scale scores improved after 12 months. Further analysis of the RBANS delayed memory domain in the placebo group suggested that the observed improvement (i.e., a 14% improvement) was driven by one subject. When this subject was removed, an improvement of 4% was denoted. As an example, Figure 3 shows the change in individual scores recorded for the immediate memory domain of the RBANS in both groups. 

### 3.4. Observed Change in Episodic Memory

Table 6 shows trends in improvements in episodic memory in the active intervention group where individuals consuming the nutritional supplement recorded fewer errors at the latter (and more challenging) stages of the paired associated learning (PAL) task in comparison to individuals in the placebo group where scores either remained unchanged or worsened. A minimal improvement (+1%) in the first attempt memory score was also observed among individuals in the active group while individuals in the placebo group declined slightly (by 1%).

### 3.5. Observed Change in Working Memory

Table 7 summaries the observed changes in working memory following the 12-month intervention period. For both groups, all tasks that assessed working memory either remained unchanged or worsened over the 12-month study period, with the exception of the number of between and total errors made at stage 8 which improved slightly (by two points each) among individuals in the active intervention group and the number of total errors made at stage 4 of the SWM task, which improved slightly (by one point) among individuals in the placebo group. Of note, individuals in the placebo group recorded significantly fewer between and total errors at stage 8 of the SWM tasks at baseline.

### 3.6. Observed Change in Reaction Time

Table 7 also shows the observed changes in reaction time following the 12-month intervention period. Results for reaction times were also mixed for both groups. Trends in improvements in simple reaction time were observed in both groups (by an average of 2 milliseconds [ms]). In addition, trends in improvements in five-choice reaction time (by an average of 20 ms) were observed among individuals in the active intervention group while declines (by an average of 12 ms) were recorded among individuals receiving placebo. Both simple and five-choice movement times declined in both groups (by an average of 13 ms and 29 ms for active and placebo groups, respectively) with overall trends suggesting a lesser decline among individuals in the active intervention group. Finally, error scores for all reaction time assessments remained unchanged with the exception of the number of errors made during the five-choice reaction time assessment, which declined (by one point) in the placebo group (see Table 7).

## 4. Discussion

Given that ω-3FAs, xanthophyll carotenoids, and vitamin E are present in brain tissue, and given their ability to attenuate mechanisms involved in the pathogenesis of AD (namely oxidative stress and neuro-inflammation), it is likely that they play an important neuroprotective role by maintaining and optimizing cognition and reducing the risk of cognitive decline. Importantly, previous studies have shown that cognitively impaired individuals are deficient in these key nutritional compounds in comparison to cognitively healthy individuals. Therefore, it is likely that specific population groups, such as individuals with MCI, will benefit from nutritional intervention. CARES Trial 1 was a parallel group, double-blind, placebo-controlled, randomized clinical trial designed to examine the effect of targeted nutritional intervention on cognitive performance among individuals with MCI. Following 12-month supplementation with a combination of ω-3FAs (DHA and EPA), xanthophyll carotenoids (L, Z and MZ), and vitamin E (α-tocopherol), this exploratory study identified trends in improved performance in episodic memory, immediate memory, attention and delayed memory among individuals with clinically confirmed MCI.

### 4.1. Significance and Interpretation of Findings

This exploratory study is the first of its kind to examine the impact of a combination of ω-3FAs, xanthophyll carotenoids, and vitamin E on cognition in individuals with MCI. Previous studies have examined the effects of varying combinations of nutritional compounds on the cognitive health of individuals with MCI (see [60] for a review). While many of the studies included in the review performed by Solfrizzi and colleagues [60] observed reductions in brain atrophy in individuals with MCI, no positive effects on cognition were found, with the exception of [61] where improvements in the dementia rating scale were reported following 6-month consumption of a combination of vitamin B_12_, α-tocopherol, s-adenosylmethionine, N-acetylcysteine and acetyl-L-carnitine. Interestingly, supplementation with ω-3FAs alone have yielded positive results among individuals with MCI. A meta-analysis of 15 interventional trials suggested a benefit of DHA supplementation in terms of improving episodic memory among mildly impaired individuals [62]. Other studies have also reported benefits in memory (episodic, short-term, working, and immediate verbal), processing speed, and attention [63,64,65] following ω-3FA supplementation. In contrast, supplementation with vitamin E is less promising as, currently, no improvements in cognitive performance have been observed in MCI samples [66] and none have examined the impact of xanthophyll carotenoid supplementation alone on cognition in individuals with MCI. Importantly, to date, none have examined the potential synergistic effect of the ω-3FAs DHA and EPA, the xanthophylls L, Z and MZ, and vitamin E in the α-tocopherol form on cognitive function among individuals with MCI.

In the present exploratory study, individuals with MCI in the active intervention group responded positively to 12-month nutritional supplementation in terms of statistically significant improvements in tissue carotenoid concentrations, as well as statistically significant increases in blood concentrations of serum L, serum MZ and plasma DHA. Of note, individuals in the active intervention group responded poorly to vitamin E (α-tocopherol) supplementation. Reasons underlying the poor vitamin E response following supplementation remain unclear. While in accordance with international recommended dietary allowances (RDAs) [67,68], it is possible that the daily dosage of vitamin E (i.e., 15 mg) used in the present study was too low (in comparison to other interventional studies (e.g., [69,70]) to have any meaningful effect.

Most of the positive outcomes identified in the present exploratory study relate to performance in tasks assessing memory. Memory deficits, which involve difficulties with the encoding, storage and retrieval of information, are commonplace among individuals with MCI. Specifically, impairment in episodic memory has been frequently documented in MCI and is an inherent feature of amnestic MCI [71,72]. Episodic memory refers to the ability to learn, store and retrieve information about experiences that occurred at a particular time and place (e.g., remembering where you parked your car in a multi-story carpark, remembering the details of a family event attended in the past few weeks [73]). Encoding, retention and retrieval difficulties are likely due to changes in the relevant neurocircuitry including frontal, temporal and medial temporal lobe regions, the hippocampus and adjacent cortical areas [74,75].

### 4.2. Neurobiological Mechanisms

The brain is a lipid-dense organ containing large amounts of ω-3FAs (and DHA in particular) [76]. Additionally, xanthophyll carotenoids selectively accumulate in brain tissue including frontal and temporal cortices. Ω-3FAs are considered to play an important role in neurological health. It has been suggested that DHA plays an important role in the control and resolution of neuro-inflammation. This role is performed by a number of pathways, including being converted into bioactive lipid metabolites such as endocannabinoid epoxides (molecules that are responsible for antiangiogenic effects, vasodilatory actions, and regulation of platelet aggregation) [77]. It has also been suggested that DHA downregulates the expression of genes involved in the synthesis of pro-inflammatory eicosanoids produced from the ω-6FA arachidonic acid [78]. Despite EPA being stored in the brain in low amounts, it has been demonstrated that this fatty acid is important for neural efficiency. This suggests that EPA may positively influence pathways that regulate high-order cognitive functions [79]. It has also been suggested that EPA can facilitate enzymatic processes required to inhibit neuronal damage from inflammation and oxidative stress [80].

Additionally, xanthophyll carotenoids are premised to be neuroprotective primarily owing to their antioxidant properties. Due to their conjugated double-bond structure, carotenoids are efficient scavengers of reactive oxygen species [81]. The lipid solubility of carotenoids also enables them to reduce the susceptibility of cellular membranes and lipoproteins to oxidative damage through free-radical scavenging [82]. L and Z have been shown to positively impact neural efficiency [83,84] and cellular communication via gap junctions [85]. Carotenoids can also combat inflammation. For example, it has been shown that carotenoids are involved in the modulation of inflammatory cells and pro-inflammatory enzymes, the downregulation of pro-inflammatory molecule production, and the attenuation of inflammatory gene expression [86].

While it cannot be asserted that improvements in specific cognitive domains (as a result of supplementation for example) necessarily negates any pre-existing risk for going on to develop AD, nevertheless it is reasonable to hypothesize that the observed improvements in the encoding and memory retrieval process among individuals consuming the nutritional intervention may reflect favorable changes in the physiological functionality, structural integrity and synaptic activity of brain regions involved in memory, and that these favorable changes may be attributable to the enrichment of the aforementioned nutritional compounds. Moreover, the observed trends in improvements in cognitive outcomes may help to favorably alter the risk profile of these individuals for further cognitive decline in the future by enriching the neurocognitive environment.

### 4.3. Strengths and Limitations

Strengths of CARES Trial 1 include a comprehensive assessment of MCI using sensitive and validated diagnostic measurement tools at screening, enrolment and follow-up assessments. Furthermore, the use of a consensus panel provided in-depth characterization of all individuals and the implementation of robust inclusion and exclusion criteria ensured a clean dataset. The interpretation, analysis and generalizability of results from CARES Trial 1 were limited due to the lack of statistical power in the trial. In order to ensure sufficient statistical power to test the proposed research hypothesis, CARES Trial 1 aimed to recruit 60 individuals with MCI. In order to achieve this target (and allowing for a 10% attrition rate), it was anticipated that a large number of individuals would have to be screened. Despite increased attempts, the identification and recruitment of individuals with MCI proved extremely challenging. The high rate (30%) of individuals who chose not to participate (despite their eligibility) and high attrition rate (as illustrated in Figure 1) were unforeseen and highlights the challenges of conducting research in the MCI population. A number of attempts were made to address the challenges of identifying and enrolling individuals with MCI. These included widening the recruitment catchment area from one city (i.e., Waterford, Ireland) to the entire South-East region of Ireland and hosting briefing meetings with the relevant consultant geriatricians and psychiatrists of old age in the region. Repeated written communication was also carried out to remind the relevant consultant geriatricians and psychiatrists of old age in the region about the clinical trial (including the project aims and inclusion criteria). Despite increased attempts, MCI baseline numbers remained low and drop-out rates remained high.

The small sample size of CARES Trial 1 also precluded the study from comparing MCI subtypes (i.e., amnestic versus non-amnestic) and examining potential relationships between nutritional status and cognitive outcomes. CARES Trial 1 may also be subject to selection bias, given that individuals were primarily recruited from the clinic setting. Finally, depressive symptoms were not assessed at screening. However, depression under active review was part of the exclusion criteria and may counteract this perceived limitation. Despite these limitations, this exploratory study provides encouraging preliminary data. We have shown that individuals with MCI respond (in tissue and in blood) to targeted nutritional supplementation. Additionally, we have observed trends in improved performance in tasks assessing episodic memory and global cognition (namely immediate memory, delayed memory and attention).

## 5. Conclusions

In conclusion, the present exploratory study has identified trends in improved performance in episodic memory and global cognition among individuals with clinically confirmed MCI following 12-month targeted nutritional supplementation with a combination of ω-3FAs, xanthophyll carotenoids, and vitamin E. Despite the heterogeneity of MCI, studying individuals with this condition provides a unique opportunity to examine the efficacy of nutrition as a preventative approach in slowing the progression of cognitive impairment and improving cognitive-related outcomes. Given that there has been little clinical success with pharmacological strategies for cognitive decline and AD and given that current thinking surrounding the amyloid hypothesis is being challenged by many in the scientific community [87], shifting focus towards preventative approaches is timely and warranted. The results of the present study are highly promising and highlight the potential of nutrition as a preventative strategy for modifying or delaying MCI progression and improving cognitive outcomes. MCI presents a unique opportunity to examine the potential of nutrition for improving cognitive outcomes in individuals at an early stage of impairment. Despite the small sample size, this exploratory interventional work has not only addressed a gap in the literature but has also shown that individuals with clinically confirmed MCI respond positively to targeted nutritional supplementation. Larger-scaled and appropriately powered interventional trials are clearly warranted to confirm this finding and to explore interactions between nutritional compounds and cognitive status.

## Figures and Tables

**Figure 1 jpm-10-00043-f001:**
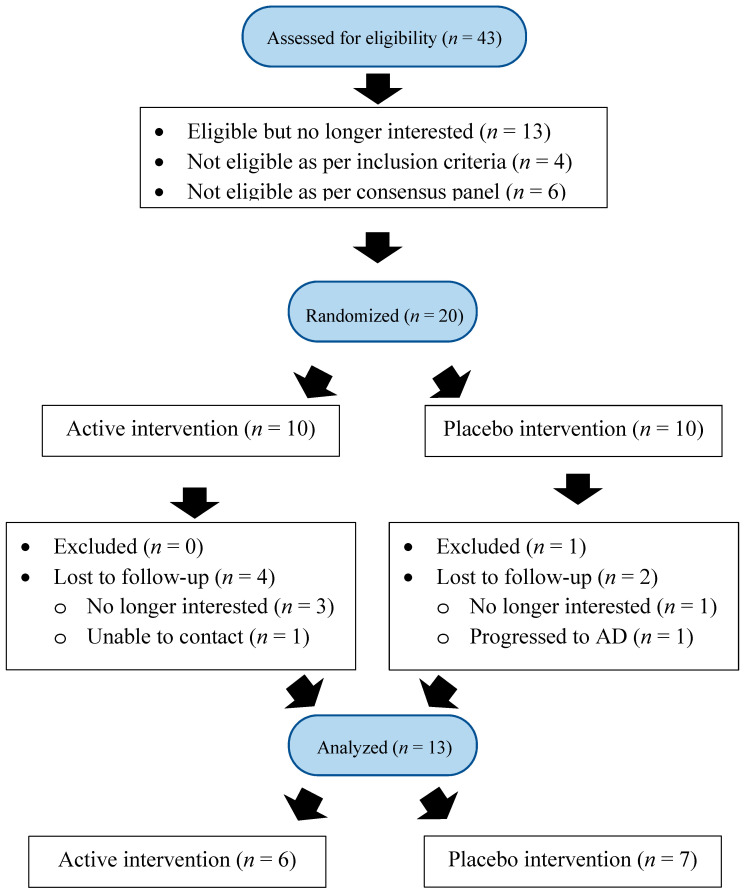
Consolidated Standards of Reporting Trials (CONSORT) flow diagram for the Cognitive impAiRmEnt Study (CARES) Trial 1.

**Figure 2 jpm-10-00043-f002:**
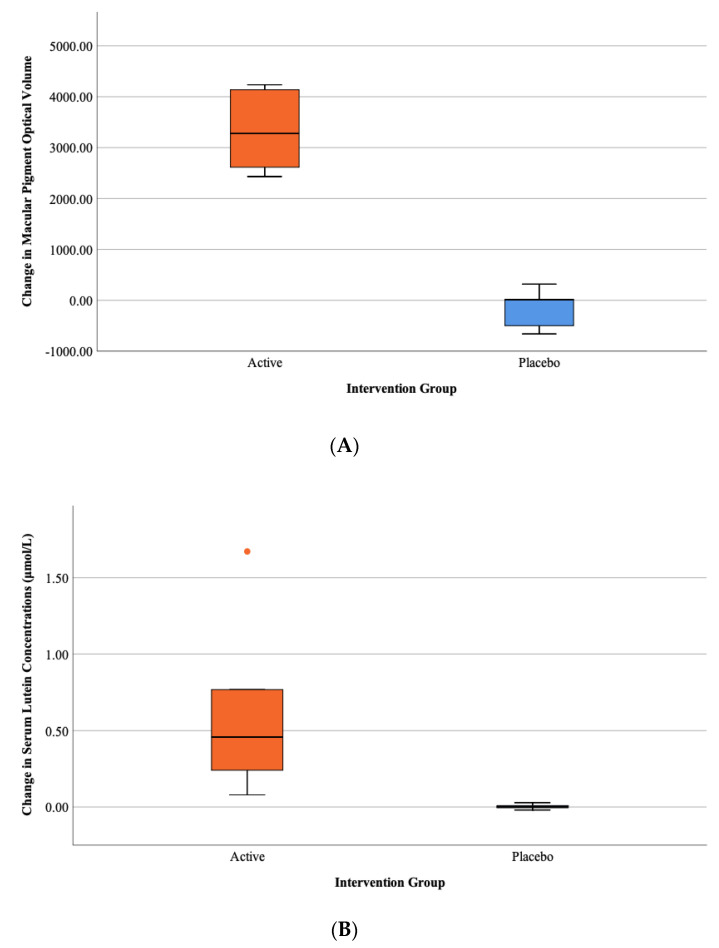
(**A**) Box plots illustrating change in MPOV over 12 months for active and placebo intervention groups. (**B**) Box plot illustrating change in serum lutein over 12 months for active and placebo intervention groups. (**C**) Box plot illustrating change in plasma DHA over 12 months for active and placebo intervention groups.

**Figure 3 jpm-10-00043-f003:**
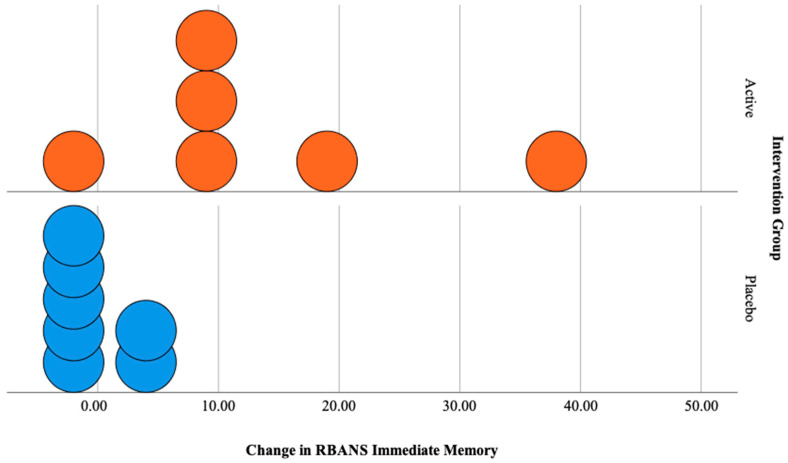
Dot plots illustrating change in individual scores recorded for the immediate memory domain of the RBANS for active and placebo intervention groups.

**Table 1 jpm-10-00043-t001:** Tasks performed in CARES to assess cognition using the Cambridge neuropsychological test automated battery (CANTAB).

Cognitive Domain	Task	Description	Outcome Measure	Desirable Score	Performance Ranges
Comprehension	MOT	Individuals must touch the flashing cross shown in different locations on the screen.	Latency (speed of response)	Lower	-
			Total correct	Higher	0–10
			Total errors	Lower	0–10
Executive function (working memory)	SWM	The aim of this test is that, by touching the boxes and using a process of elimination, the individual should find one ‘token’ in each of the boxes and use them to fill up an empty column on the right-hand side of the screen. The key task instruction is that the computer will never hide a token in the same coloured box, so once a token is found in a box the individual should not return to that box to look for another token.	Between errors	Lower	0–90
			Total errors	Lower	0–90
			Strategy	Lower	2–14
Reaction time	RTI	Individuals must press and hold down a touchscreen button at the bottom of the screen. Circles are presented at the top of the screen (one for simple mode, and five for the five-choice mode). In each case, a yellow spot will appear in one of the circles. Individuals must respond to the spot as quickly as they can by letting go of the button and touching inside the circle where the yellow spot appeared.	Simple reaction time	Lower	100–5100
			Simple movement time	Lower	100–5100
			Simple error score	Lower	0–30
			Five-choice reaction time	Lower	100–5100
			Five-choice movement time	Lower	100–5100
			Five-choice error score	Lower	0–30
Episodic memory	PAL	Boxes are displayed on the screen and open one by one in a randomized order to reveal patterns hidden inside. The patterns are then displayed in the middle of the screen, one at a time, and the individual must touch the box where the pattern was originally located. If the individual makes an error, the patterns are re-presented to remind the individual of their locations.	First attempt memory score	Higher	0–20
			No. patterns reached	Higher	2–8
			Total errors adjusted	Lower	0–70

Performance ranges: the minimum and maximum possible score for each measure; Latency: the speed (milliseconds) of response to the stimulus; Total correct: the number of correct responses; Total errors: the distance between the center of the cross and the location touched; Between errors: the number of times the individual revisits a box in which a token has previously been found; Total errors: the number of times a box is selected that is certain not to contain a token and therefore should not have been visited by the individual; Strategy: for problems with six boxes or more. The number of distinct boxes used by the individual to begin a new search for a token, with the same problem; Simple reaction time: the duration between the onset of the stimulus and the time at which the individual released the button. Calculated for correct trials, where the stimulus could appear in one location only; Simple movement time: the time taken to touch the stimulus after the button has been released. Calculated for correct trials, where the stimulus could appear in one location only; Simple error score: the number of trials where the response status is any error (i.e., an inaccurate response, no response, or a premature response) for the assessment trial where stimuli appear on one location only. The error may be an inaccurate response, no response, or a premature response; Five-choice reaction time: the duration between the onset of the stimulus and the release of the button. Calculated for correct, assessed trials where the stimulus could appear in any of the five locations; Five-choice movement time: the time taken to touch the stimulus after the button has been released. Calculated for correct, assessed trials where the stimulus could appear in any of the five locations; Five-choice error score: the number of trials where the response status is any error (i.e., an inaccurate response, no response, or a premature response) for assessment trials where stimuli appear in any of five locations; First attempt memory score: the number of correct box choices that were made on the first attempt during assessment problems; No. patterns reached: the number of patterns on the last problem reached in the task; Total errors adjusted: the number of times the individual chose the incorrect box for a stimulus on assessment problems, plus an adjustment for the estimated no. errors they would have made on any problems, attempts and recalls they did not reach.

**Table 2 jpm-10-00043-t002:** Baseline demographic, health and lifestyle data of active and placebo intervention groups.

Variable	Active (*n* = 10) Median (IQR)	Placebo (*n* = 9) Median (IQR)	Sig.
*Demographic data*			
Age (years)	73.5 (69.5–80.5)	72.0 (69.5–75.5)	0.549
Sex ([*n*]; [% female])	5 (50.0%)	8 (88.9%)	0.069
Education (years)	17.5 (15.5–21.0)	15.0 (15.0–16.5)	0.095
*Health and lifestyle data*			
Medications	6.0 (3.0–8.3)	5.0 (2.0–5.5)	0.133
Exercise (min/week)	217.0 (0–326.3)	210.0 (45.0–375.0)	0.720
Smoking ([*n*]; [%])			0.463
Never	5 (50%)	6 (66.7%)	
Past	5 (50%)	3 (33.3%)	
Current	0	0	
Alcohol consumption ([*n*]; [%])			0.473
0 units	5 (50.0%)	4 (44.4%)	
1 unit	3 (30.0%)	2 (22.2%)	
2–5 units	0	2 (22.2%)	
6–10 units	2 (20.0%)	1 (11.2%)	
>10 units	0	0	
BMI (kg/m^2^)	26.3 (25.8–30.7)	26.7 (25.1–27.8)	0.720
*Nutritional status*			
MPOV	6987 (2969–9080)	4682 (3740–7311)	0.497
SCS	4.20 (3.75–5.60)	4.65 (4.23–5.40)	0.541
Serum L	1.72 (1.03–2.21)	1.32 (1.04–1.86)	0.815
Serum Z	1.34 (1.06–1.55)	2.03 (1.14–2.12)	0.236
Serum MZ	2.50 (1.85–3.50)	2.15 (1.48–3.78)	0.743
Serum vitamin E	11.0 (8.10–15.55)	8.65 (7.33–11.80)	0.236
Plasma DHA	0.80 (0.55–5.10)	0.45 (0.30–1.10)	0.236
Plasma EPA	1.52 (0.97–2.15)	1.08 (0.95–1.84)	0.541
Folate	12.10 (10.85–14.20)	11.90 (10.88–13.78)	0.888
Vitamin B_12_	14.10 (12.55–15.15)	12.90 (12.48–13.38)	0.236

Data displayed are median (inter quartile range) for numeric data and actual number and percentages for categorical data; Education: age (years) completed formal education; Medications: the number of prescribed medications consumed; Alcohol consumption: units/week: MPOV: a volume of macular pigment calculated as macular pigment average times the area under the curve out to 7° eccentricity (measured using the Heidelberg Spectralis^®^). SCS: skin carotenoid score (measured using the Pharmanex BioPhotonic Scanner). Serum lutein, zeaxanthin, *meso*-zeaxanthin and vitamin E concentrations are expressed in µmol/L; Plasma docosahexaenoic acid and eicosapentaenoic acid concentrations expressed in µmol/L; Serum folate concentrations expressed in ng/mL; Serum vitamin B_12_ concentrations expressed in pg/mL. Serum carotenoid and vitamin E data were not available for two individuals in the active intervention and one individual in the placebo group. Plasma DHA and EPA were not available for one individual in the active intervention and one individual in the placebo group.

**Table 3 jpm-10-00043-t003:** Baseline cognitive function and functional ability data of active and placebo intervention groups.

Variable	Active (*n* = 10) Median (IQR)	Placebo (*n* = 9) Median (IQR)	Sig.
*Global cognition*			
MoCA	21.0 (18.8–24.0)	21.0 (19.0–24.0)	0.842
RBANS immediate memory	78.0 (64.0–82.5)	85.0 (72.5–98.5)	0.156
RBANS visuospatial	101.0 (92.0–109.0)	100.0 (91.5–110.5)	0.842
RBANS language	89.0 (84.5–93.0)	88.0 (82.0–92.0)	0.661
RBANS attention	81.0 (71.0–98.5)	79.0 (77.0–89.5)	0.905
RBANS delayed memory	75.0 (47.0–87.5)	71.0 (58.0–87.0)	<0.999
RBANS total scale	78.0 (75.0–84.0)	82.0 (73.5–85.0)	0.497
4 mountains test	6.5 (6.0–7.8)	7.0 (5.5–7.5)	0.905
*Comprehension*			
Latency	1165.9 (812.9–1322.2)	1152.8 (991.75–1301.5)	0.842
Total errors	0	0	0.720
*Working memory*			
Between errors			
Stage 4	1.0 (0.8–2.0)	(1.0–2.5)	0.497
Stage 6	6.0 (4.3–7.5)	6.0 (6.0–7.0)	0.497
Stage 8	16.0 (15.0–17.0)	12.5 (11.0–14.0)	0.006
All stages	21.5 (21.0–26.5)	20.0 (18.0–22.5)	0.182
Total errors			
Stage 4	(0.8–2.0)	2.0 (1.0–2.5)	0.447
Stage 6	6.0 (4.3–7.5)	6.0 (6.0–7.0)	0.497
Stage 8	16.5 (15.0–17.3)	13.0 (11.0–14.8)	0.012
All stages	21.5 (21.0–28.3)	22.0 (18.0–22.5)	0.447
Strategy	10.0 (9.0–11.3)	9.0 (8.5–10.0)	0.315
*Reaction time*			
Simple reaction time	382.4 (370.7–463.6)	372.9 (354.9–411.5)	0.356
Simple movement time	296.5 (249.6–336.6)	318.9 (277.3–352.3)	0.497
Simple error score	2.0 (0.8–3.5)	1.0 (0–3.0)	0.400
Five-choice reaction time	458.1 (415.7–510.3)	435.1 (406.5–485.5)	0.549
Five-choice movement time	310.3 (286.5–367.0)	313.0 (294.2–335.4)	<0.999
Five-choice error score	0.5 (0–1.3)	1.0 (0–2.0)	0.549
*Episodic memory*			
First attempt memory score	4.0 (3.5–5.5)	4.0 (2.5–6.5)	0.905
No. patterns reached	6.0 (4.0–6.5)	6.0 (4.0–6.0)	0.549
Total errors adjusted stage 2	0 (0–0.3)	0 (0–0.5)	0.905
Total errors adjusted stage 4	6.0 (2.3–8.8)	6.0 (0.5–9.5)	0.780
Total errors adjusted stage 6	18.5 (12.3–20.0)	20.0 (14.5–20.0)	<0.999
Total errors adjusted stage 8	28.0 (25.3–28.0)	28.0 (28.0–28.0)	0.720
Total errors adjusted all stages	52.5 (42.8–59.0)	48.0 (45.0–58.0)	0.780
*Functional ability*			
BADLS	20.0 (15.5–20.0)	20.0 (20.0–20.0)	0.243
AQ	8.0 (4.8–13.8)	5.0 (2.5–8.0)	0.113

Data displayed are median (inter quartile range); MoCA: Montreal Cognitive Assessment; RBANS: Repeatable Battery for the Assessment of Neuropsychological Status; BADLS: Bristol Activities of Daily Living Scale; AQ: Alzheimer’s Questionnaire.

**Table 4 jpm-10-00043-t004:** Change in nutritional status over 12 months between active and placebo intervention groups.

Variable	Active Intervention					Placebo Intervention					
	*n*	Baseline Median (IQR)	12 Months Median (IQR)	%Δ	Outcome	*n*	Baseline Median (IQR)	12 Months (Median (IQR)	%Δ	Outcome	Sig.
*Nutrition*											
MPOV	6	6987 (2947–9080)	10,363 (5488–12,906)	+62	Improved	7	4682 (3838–7264)	4300 (3827–7277)	−2	Declined	**0.001**
SCS	6	27,250 (18,250–36,705)	37,000 (30,000–60,250)	+79	Improved	7	17,000 (15,000–35,000)	21,000 (14,000–38,000)	+2	Improved	**0.014**
Serum L	5	0.152 (0.107–0.217)	0.562 (0.339–1.388)	+421	Improved	7	0.104 (0.067–0.188)	0.133 (0.067–0.168)	+5	Improved	**0.003**
Serum Z	5	0.059 (0.036–0.078)	0.075 (0.056–0.125)	+58	Improved	7	0.037 (0.035–0.051)	0.042 (0.037–0.049)	+1	Improved	0.247
Serum MZ	5	0	0.068 (0.031–0.234)	-	Improved	7	0	0	0	Unchanged	**0.003**
Serum vit. E	5	23.988 (20.346–30.264)	23.015 (21.010–31.802)	−4	Declined	7	24.426 (22.007–26.274)	24.704 (23.074–26.093)	+1	Improved	>0.999
Plasma DHA	5	235.730 (153.115–258.315)	291.910 (262.170–406.650)	+59	Improved	7	200.480 (165.230–212.230)	234.630 (204.150–267.680)	+17	Improved	**0.048**
Plasma EPA	5	164.500 (90.000–199.070)	178.410 (178.410–205.425)	+6	Improved	7	129.530 (127.550–146.620)	147.020 (108.080–201.850)	+13	Improved	0.639

Data displayed are median (inter quartile range); %Δ: average percentage change per subject; Sig.: Level of significance of the observed change over time between active and placebo intervention groups; MPOV, a volume of macular pigment calculated as macular pigment average times the area under the curve out to 7° eccentricity (measured using the Heidelberg Spectralis^®^); SCS: skin carotenoid score (obtained using the Pharmanex BioPhotonic Scanner); Serum lutein, zeaxanthin, *meso*-zeaxanthin and vitamin E concentrations are expressed in µmol/L; Plasma docosahexaenoic acid and eicosapentaenoic acid concentrations expressed in µmol/L.

**Table 5 jpm-10-00043-t005:** Average percentage change in global cognition over 12 months between active and placebo intervention groups.

Variable	Active Intervention					Placebo Intervention				
	*n*	Baseline Median (IQR)	12 Months Median (IQR)	%Δ	Outcome	*n*	Baseline Median (IQR)	12 Months (Median (IQR)	%Δ	Outcome
*Global cognition* (*RBANS*)										
Immediate memory	6	78.0 (73.3–82.5)	91.0 (81.3–100.8)	+18	Improved	7	94.0 (85.0–100.0)	90.0 (81.0–103.0)	0	Unchanged
Visuospatial	6	107.0 (101.5–110.8)	105.0 (101.5–112.0)	−1	Declined	7	96.0 (87.0–109.0)	96.0 (84.0–109.0)	0	Unchanged
Language	6	91.0 (88.0–96.0)	89.0 (88.0–93.0)	−1	Declined	7	88.0 (82.0–92.0)	92.0 (72.0–105.0)	+2	Improved
Attention	6	84.5 (68.0–103.8)	91.0 (79.0–100.0)	+7	Improved	7	79.0 (75.0–85.0)	79.0 (79.0–94.0)	0	Unchanged
Delayed memory	6	85.0 (63.3–96.5)	86.0 (63.0–106.3)	+12	Improved	7	71.0 (60.0–93.0)	90.0 (78.0–98.0)	+14	Improved
Total scale	6	82.0 (78.0–91.5)	87.0 (82.3–99.3)	+6	Improved	7	82.5 (76.3–87.3)	88.5 (78.5–91.3)	+3	Improved

Data displayed are median (inter quartile range); %Δ: average percentage change per subject; RBANS: Repeatable Battery for the Assessment of Neuropsychological Status.

**Table 6 jpm-10-00043-t006:** Average change per subject in episodic memory, working memory and reaction time over 12 months between active and placebo intervention groups.

Variable	Active Intervention					Placebo Intervention				
	*n*	Baseline Median (IQR)	12 Months Median (IQR)	Δ	Outcome	*n*	Baseline Median (IQR)	12 Months (Median (IQR)	Δ	Outcome
*Episodic memory* (*PAL*)										
First attempt memory score	6	4.5 (3.3–7.5)	6.5 (2.0–9.8)	+1	Improved	7	4.0 (3.0–6.0)	3.0 (2.0–6.0)	−1	Declined
Total errors adjusted stage 2	6	0 (0–1.3)	0 (0)	−1	Improved	7	0 (0)	0 (0–2.0)	−1	Improved
Total errors adjusted stage 4	6	6.0 (2.3–8.0)	7.5 (2.0–11.8)	+2	Declined	7	6.0 (0–10.0)	8.0 (4.0–10.0)	+2	Declined
Total errors adjusted stage 6	6	15.0 (7.0–18.5)	14.0 (5.8–20.0)	−1	Improved	7	20.0 (15.0–20.0)	20.0 (15.0–20.0)	0	Unchanged
Total errors adjusted stage 8	6	28.0 (15.3–28.0)	20.5 (8.8–28.0)	−4	Improved	7	28.0 (28.0–28.0)	28.0 (28.0–28.0)	0	Unchanged
Total errors adjusted all stages	6	54.5 (24.8–54.5)	39.5 (17.8–59.8)	−4	Improved	7	48.0 (47.0–58.0)	51.0 (48.0–58.0)	+2	Declined

Data displayed are median (inter quartile range); Δ: average change per subject; PAL: paired associated learning; First attempt memory score: the number of correct box choices that were made on the first attempt during assessment problems; Total errors adjusted: the number of times the individual chose the incorrect box for a stimulus on assessment problems, plus an adjustment for the estimated no. errors they would have made on any problems, attempts and recalls they did not reach.

**Table 7 jpm-10-00043-t007:** Average change per subject in working memory and reaction time over 12 months between active and placebo intervention groups.

Variable	Active Intervention					Placebo Intervention				
	*n*	Baseline Median (IQR)	12 Months Median (IQR)	Δ	Outcome	*n*	Baseline Median (IQR)	12 Months (Median (IQR)	Δ	Outcome
*Working memory* (*SWM*)										
Between errors stage 4	6	1.0 (0.8–2.0)	1.5 (0.8–2.5)	+1	Declined	7	2.0 (1.0–3.0)	2.0 (0–3.0)	0	Unchanged
Between errors stage 6	6	5.5 (2.0–7.5)	6.0 (5.5–8.8)	+2	Declined	7	6.0 (6.0–7.0)	9.0 (6.0–10.0)	+1	Declined
Between errors stage 8	5	17.0 (15.0–17.0)	16.0 (10.0–18.0)	−2	Improved	5	12.0 (10.5–15.5)	13.0 (12.5–18.0)	+2	Declined
Between errors all stages	5	21.0 (20.0–24.0)	26.0 (15.5–27.0)	−2	Improved	5	20.0 (19.0–23.5)	21.0 (20.0–24.0)	+2	Declined
Total errors stage 4	6	1.0 (0.8–2.0)	1.5 (0.8–2.5)	+1	Declined	7	2.0 (1.0–3.0)	2.0 (0–3.0)	−1	Improved
Total errors stage 6	6	5.5 (2.0–7.5)	6.0 (5.5–9.0)	+2	Declined	7	6.0 (6.0–7.0)	10.0 (6.0–11.0)	+2	Declined
Total errors stage 8	5	17.0 (15.0–18.5)	16.0 (10.5–19.0)	−2	Improved	5	12.0 (10.5–16.5)	15.0 (12.5–18.0)	+2	Declined
Total errors all stages	5	21.0 (20.0–25.5)	26.0 (16.0–28.0)	0	Unchanged	5	22.0 (13.5–22.8)	22.5 (20.5–24.3)	+4	Declined
Strategy	5	10.0 (8.5–12.0)	10.0 (8.5–11.5)	0	Unchanged	5	9.0 (6.8–10.5)	9.0 (9.0–10.5)	+1	Declined
*Attention* (*RTI*)										
Simple reaction time	6	381.1 (364.4–463.6)	397.3 (342.8–453.9)	−2	Improved	7	389.7 (340.4–430.6)	369.2 (355.3–448.7)	−2	Improved
Simple movement time	6	302.9 (181.9–432.5)	316.0 (244.7–433.5)	+13	Declined	7	318.9 (274.6–337.7)	344.2 (246.3–449.8)	+29	Declined
Simple error score	6	1.5 (0–2.8)	1.0 (0–2.5)	0	Unchanged	7	1.0 (0–3.0)	1.0 (1.0–3.0)	0	Unchanged
Five-choice reaction time	6	483.7 (420.5–519.5)	445.7 (429.1–477.4)	−20	Improved	7	435.1 (391.2–479.0)	450.7 (393.4–488.3)	+12	Declined
Five-choice movement time	6	316.6 (279.2–443.8)	333.2 (261.8–417.4)	+13	Declined	7	310.5 (289.1–320.0)	332.8 (323.3–416.4)	+29	Declined
Five-choice error score	6	1.0 (0–2.3)	0 (0–3.3)	0	Unchanged	7	1.0 (0–2.0)	1.0 (0–3.0)	+1	Declined

Data displayed are median (inter quartile range); Δ: average change per subject. SWM: spatial working memory; Between errors: the number of times the individual revisits a box in which a token has previously been found; Total errors: the number of times a box is selected that is certain not to contain a token and therefore should not have been visited by the individual; Strategy: for problems with six boxes or more. The number of distinct boxes used by the individual to begin a new search for a token, with the same problem; RTI: reaction time; Simple reaction time: the duration between the onset of the stimulus and the time at which the individual released the button. Calculated for correct trials, where the stimulus could appear in one location only; Simple movement time: the time taken to touch the stimulus after the button has been released. Calculated for correct trials, where the stimulus could appear in one location only; Simple error score: the number of trials where the response status is any error (i.e., an inaccurate response, no response, or a premature response) for the assessment trial where stimuli appear on one location only. The error may be an inaccurate response, no response, or a premature response; Five-choice reaction time: the duration between the onset of the stimulus and the release of the button. Calculated for correct, assessed trials where the stimulus could appear in any of the five locations; Five-choice movement time: the time taken to touch the stimulus after the button has been released. Calculated for correct, assessed trials where the stimulus could appear in any of the five locations; Five-choice error score: the number of trials where the response status is any error (i.e., an inaccurate response, no response, or a premature response) for assessment trials where stimuli appear in any of five locations.

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
