# Peer review of "Targeted Nutritional Intervention for Patients with Mild Cognitive Impairment: The Cognitive impAiRmEnt Study (CARES) Trial 1"

_jpm, 2020, doi:10.3390/jpm10020043_

Round 1
Reviewer 1 Report
The authors of the study wanted to examine is dietary supplementation with omega-3 and carotenoid beneficial for patients with mild cognitive impairment. The patients received supplementation or placebo for 12 months. Before onset of supplementation and after its end numerous data were collected by performing cognitive tests and questionnaires, while nutritional and biochemical status was determined by analysis in tissues and plasma/serum. The authors found increased carotenoids in tissue and blood, increased omega-3 and improvement in episodic memory and global cognition in supplemented group. However, the authors stated that small number of N is limitation of the study.
This is interesting study, but there are few points of concern:
Abstract:
Abstract can be written better and more fluid. Expression “selectively accumulate” and “positioned” doesn’t sound right, as omega-3 are regular parts of membrane and has numerous roles. In the line 28 there are 19 individuals, though we can see below that N= 13 .
Introduction:
-Introduction is comprehensive, but too long, it should be shorten for few sentences.
- Figure 1 is not necessary as the authors anyway don’t analyze subgroups of MCI.
-In last paragraphs of Introduction and in Materials and Methods the authors explains that novelty of the study is potential synergistic effect of omega-3, carotenoid and Vit E. However, in the Abstract Vit E, as a part of treatment, is not even mentioned. Also, this was a place to mention their previous study (2018) in which they found that combination of carotenoids and omega-e have better effect in AD patients than carotenoids alone, a this fact would support experimental design.
-Please put reference for the sentence in line 60-61.” Posterior cingulate” applies to cortex?
Material and Methods
-It is nicely written, with lot of details, but I am confused about capsules-are they commercial or specifically made for the experiment; who is omega-3 oil manufacture; is so high ratio of DHA:EPA characteristic of natural fish oils; is one capsules with O-3 oil and the other with carotenoids and Vit E, or Vit E is just there as antioxidants to protect oils?
Results
-This section is not easy to follow. Is it possible to have some subtitles to separate different results? Also, can you please, put appropriate Fig number somewhere at the beginning of the text, as for example, tags for Table and Fig 3 and Fig 6 are at the end of the text.
-Table 2A has N=10 and 9, and table 3 N=6 and 7.
-Abbreviation SWM is explained in line 712, thou it was mentioned for the first time earlier (line 454(
-What is difference between tables 2b and 3? Both of them contain baseline values of active and placebo group, but Table 2b has N=9 and 10 and Table 3 has N=6 and 7 and, some values are same in both tables and some are different.
Generally, the main flow of this study is small number of individuals observed over a long period where question of adherence to dietary supplementation is raised. So this can be only preliminary study.
Author Response
"Please see the attachment".

Reviewer 2 Report
The report "Targeted nutritional intervention for patients with 3 mild cognitive impairment: the cognitive 4 impAiRmEnt Study (CARES) Trial 1" by Dr. Power et al., is a poor study with nineteen individuals.
Authors report that the study had identified trends in improved performance in episodic memory and global cognition in MCI patients. The heterogeneity of the patients is not good enough.
Something more to obtain better results will by to try biochemistry, western-blot technique, some oxidative stress determination, RT-PCR, and so on.
In summary, even though the study seems relatively methodological sound, it provides little or no enough information. I do not find it publishable in this journal at this moment, try to improve the paper to give you a positive evaluation
Author Response
"Please see the attachment".

Round 2
Reviewer 1 Report
The authors improved manuscripts and I think it is suitable for publishing.
Reviewer 2 Report
Although the study is still poor regarding the number of patients, this referee accepts the claims indicated by the authors. In addition, the impact of the journal allows this level of research articles.